# Application of DNA Replicons in Gene Therapy and Vaccine Development

**DOI:** 10.3390/pharmaceutics15030947

**Published:** 2023-03-15

**Authors:** Kenneth Lundstrom

**Affiliations:** PanTherapeutics, CH1095 Lutry, Switzerland; lundstromkenneth@gmail.com

**Keywords:** DNA replicons, self-replication RNA viruses, RNA amplification, alphaviruses, flaviviruses, cancer gene therapy, vaccine development

## Abstract

DNA-based gene therapy and vaccine development has received plenty of attention lately. DNA replicons based on self-replicating RNA viruses such as alphaviruses and flaviviruses have been of particular interest due to the amplification of RNA transcripts leading to enhanced transgene expression in transfected host cells. Moreover, significantly reduced doses of DNA replicons compared to conventional DNA plasmids can elicit equivalent immune responses. DNA replicons have been evaluated in preclinical animal models for cancer immunotherapy and for vaccines against infectious diseases and various cancers. Strong immune responses and tumor regression have been obtained in rodent tumor models. Immunization with DNA replicons has provided robust immune responses and protection against challenges with pathogens and tumor cells. DNA replicon-based COVID-19 vaccines have shown positive results in preclinical animal models.

## 1. Introduction

Plasmid DNA-based delivery of transgenes has been used for gene therapy [1], which to a large extent has involved cancer therapy [2]. Both prophylactic and therapeutic efficacy have been achieved, and these have been appropriate in the areas of cancer therapy [3], cancer immunotherapy [4], and cancer vaccines [5]. In cancer therapy, the aim is to overexpress toxic, antitumor, or suicide genes to kill tumor cells, leading to tumor eradication and, in the best-case scenario, cure [3]. This approach has often been aided by the utilization of tumor-specific promoters [6]. In cancer immunotherapy, immunostimulatory factors such as cytokines and chemokines are overexpressed to strengthen the immune system weakened by the presence of tumors [7]. In the case of cancer vaccines, immunogenic proteins or their epitopes targeting cancer cells are introduced into plasmid DNA vectors for immunization studies in animal models and cancer patients [8]. Moreover, DNA vaccines against infectious diseases have also been engineered [9]. Similarly, surface proteins or their epitopes have been used as antigens for induction of antibody responses and protection against challenges with pathogenic viruses, bacteria, protozoans, and parasites. 

Plasmid DNA has largely been administered as intramuscular, intradermal, and subcutaneous injections [10,11]. The easy handling and rapid and inexpensive manufacturing of plasmid DNA have made DNA-based approaches attractive alternatives. However, in contrast to mRNA-based gene delivery, plasmid DNA needs to translocate to the nucleus for mRNA transcription and transport to the cytoplasm before translation can occur [12], which has hampered the efficacy of delivery and transgene expression. To address delivery issues, various technologies, such as electroporation [13], gene gun [14], and liposome [15] and polymer-based [16] nanoparticle formulations, have been applied. Especially lipid nanoparticles (LNPs) have demonstrated enhanced delivery capacity, improved transgene expression, and superior therapeutic efficacy compared to naked plasmid DNA. An alternative approach to generate improved transgene expression has been to apply DNA vectors based on self-replicating RNA viruses, which have been named DNA replicon vectors [17]. In this review, the focus is solely on DNA replicon vectors and their utilization for therapeutic interventions and vaccine development. 

## 2. DNA Replicon Vectors

Self-replicating RNA viruses including alphaviruses, flaviviruses, measles viruses, and rhabdoviruses have been frequently used for cancer therapy and vaccine development [18]. In the majority of studies, however, recombinant particles or RNA replicons have been utilized. In the case of cancers, oncolytic viruses, which can specifically target and kill tumors cells without causing any major damage to normal tissue have been applied [19]. 

The self-replicating vector systems contain the genes for the viral non-structural proteins and the gene of interest (GoI), which are in vitro transcribed from an SP6 or T7 RNA polymerase promoter [20]. Electroporation or transfection of in vitro transcribed RNA leads to extensive RNA replication in the cytoplasm of host cells, and when helper RNA containing the viral structural genes is co-administered, recombinant viral particles are generated. Replacement of the RNA polymerase promoter with a mammalian host cell compatible eukaryotic RNA polymerase type II promoter such as CMV [21] allows for direct transfection of DNA replicons. Additionally, application of DNA-based helper vectors or full-length DNA-based viral vector genome constructs allows for the generation of recombinant viral particles. Alphavirus DNA replicons contain the viral non-structural protein genes (nsP1-4), the 26S subgenomic promoter, the GoI, and a polyadenylation signal (pA) (Figure 1A) [20]. Among flaviviruses, Kunjin virus (KUN) has been used for engineering DNA replicon vectors (Figure 1B) [22]. In the KUN vector, the GoI is inserted downstream of the first 20 amino acids of the KUN C protein and upstream of the last 22 amino acids of the KUN E protein. Introduction of the foot-and-mouth disease virus 2A protease (FMDV-2A) in the KUN vector removes all KUN sequences from the recombinant protein of interest [23]. 

Although DNA-based systems provide superior safety levels compared to the application of recombinant particles, the inferior delivery compared to viral vectors has had a negative impact on their application (Figure 2). The need of nuclear delivery of DNA has further hampered the efficacy, which has been addressed by engineering of a nuclear localization signal (NLS) in the vector to facilitate the transfer of DNA to the nucleus [24]. Another issue relates to the potential integration of plasmid DNA into the host genome, which was investigated by intramuscular administration of plasmid DNA expressing the luciferase (Luc) reporter gene into mice [25]. Luc gene expression was visible in mouse skeletal muscle for more than 19 months, but the DNA was only present as extrachromosomal plasmid DNA. Although random chromosomal integration of plasmid DNA was discovered in another study [26], the frequency was very low, and it occurred when the intramuscular injection was followed by electroporation. Other studies have demonstrated only minor spread of plasmid DNA to other organs and no genomic integration [27]. Furthermore, it has been confirmed that neither anti-DNA antibodies [28] nor prokaryotic antibiotic resistance genes [29] have been detected after repeated intramuscular administration in primates.

Regarding self-replicating RNA viruses, alphaviruses [20], and flaviviruses [30] possess a positive strand ssRNA genome, which allows direct translation of their genome in infected host cells. In contrast, the RNA genome of measles viruses [31] and rhabdoviruses [32] is of negative polarity, which requires an RNA template before mRNA translation can be initiated. Although the mechanism of RNA self-replication has been described in detail previously [20], a brief presentation below and illustration in Figure 2 are included. Translation of viral RNA of the non-structural protein genes nsP1-4 generates the replicase complex responsible for highly efficient RNA replication. It is estimated that up to 10^6^ copies of subgenomic RNA are accumulated per cell leading to massive transgene expression. 

## 3. DNA Replicons and Infectious Diseases

In the context of DNA replicon vectors mainly alphaviruses have been utilized for vaccine development against infectious diseases (Table 1). Among alphaviruses, Semliki Forest virus (SFV), Sindbis virus (SIN), and Venezuelan equine encephalitis virus (VEE) DNA replicons have been used. For example, SIN DNA replicons expressing the herpes simplex virus-1 glycoprotein B (HSV-1-gB) elicited virus-specific antibodies and cytotoxic T-cells in immunized mice and provided protection against lethal challenges with HSV-1 [33]. Furthermore, a single intramuscular immunization with SIN-HSV-1-gB protected mice from lethal challenges with HSV-1. In another application of SIN DNA replicons, the measles virus (MV) hemagglutinin (pMSIN-H) and the fusion of hemagglutinin and the MV F fusion protein (pMSINH-FdU) have been administered to cotton rats [34]. Injection of pMSIN-H provided 100% protection against pulmonary measles, whereas protection with pMSINH-Fdu administration was achieved only after a booster vaccination with a live MV vaccine [34]. Moreover, SFV DNA replicons have been subjected to several studies in preclinical animal models. For example, SFV DNA replicons expressing the bovine viral diarrhea virus (BVDV) p80 (NS3) were administered to the quadricep muscles of BALB/c mice, which elicited statistically significant cytotoxic T-cell (CTL) responses and cell-mediated immune (CMI) responses against BVDV [35]. In a comparative study, SFV DNA replicons expressing the classical swine fever virus (CSFV) E2 glycoprotein and an adenovirus expressing CSFV E2 showed both strong immune responses in a pig model [36]. A heterologous vaccination regimen of prime immunization with pSFV1CS-E2 followed by rAdV-E2 booster vaccination induced significantly higher CSFV-specific neutralizing antibody titers compared to two immunizations with rAdV-E2. Furthermore, the pSFV1CS-E2/rAdV-E2 approach prevented viremia and clinical symptoms in pigs, which was not the case for the homologous immunization strategy [36]. In another prime-boost approach, efficient priming of a low dose of only 0.2 μg of SFV DNA replicons expressing the HIV Env protein and a Gag-Pol-Nef fusion protein was achieved in combination with the poxvirus Ankara (MVA) strain expressing the same HIV proteins in mice, resulting in greatly enhanced immune responses [37]. Moreover, prime-boost vaccination with different SFV DNA replicons expressing the Core, E1, E2, or non-structural p7-NS2-NS3 of the hepatitis C virus (HCV) followed by administration of the MVA strain expressing nearly the full-length HCV genome elicited high levels of HCV-specific CTL responses and humoral immune responses in mice [38]. Alphavirus DNA replicons have also been employed for the expression of the Ebola virus (EBOV) glycoprotein (GP) gene for co-expression of the GP and EBOV VP40 genes of Sudan or Zaire EBOV strains, which elicited both binding and neutralizing antibodies in mice [39]. The antibodies also showed cross-reactivity against another EBOV strain. Moreover, SFV DNA replicon vaccines showed superior immunogenicity compared to a recombinant MVA vaccine. SFV DNA replicon-based co-expression of EBOV GP and VP40 induced significantly stronger EBOV-specific humoral and cellular responses than either EBOV GP or VP40 alone in mice [40]. In another approach, a DNA replicon-based vaccine pMG4020 expressing the full-length rearranged genome of the V4020 virus provided protection against challenges with VEE in mice [41]. Moreover, immunization with the pMG4020 DNA replicon resulted in protection of rhesus macaques against VEE challenges [42]. More recently, SFV DNA replicons have been subjected to the expression of the full-length SARS-CoV-2 Spike (S) protein (DREP-S) or the S protein ecto-domain stabilized in a prefusion conformation (DREP-S^ecto^) [43]. Both DREP-S and DREP-S^ecto^ induced binding and neutralizing antibodies. Superior vaccine potency was obtained for DREP-S, eliciting high titers of SARS-CoV-2-specific IgG antibodies in mice receiving a single injection [43].

Bacterial infections and toxins have also been targeted with DNA replicons. For example, SIN DNA expressing the *Mycobacterium tuberculosis* p85 antigen elicited strong p85-specific immune responses in mice [44]. Moreover, immunized mice showed long-term protection against challenges with lethal doses of *M. tuberculosis*. Application of VEE DNA replicons for the expression of a fusion protein of the *M. tuberculosis* α-crystallin (Acr) and Ag85B antigens induced strong CD4^+^ and CD8^+^ T-cell responses in mice and prevented bacterial growth in lungs and spleen of mice after challenge with aerosol-based *M. tuberculosis* [45]. In the context of *Clostridium botulinum* neurotoxin, the serotype A (BoNT/A) Hc gene was co-expressed with the granulocyte-macrophage colony-stimulating factor (GM-CSF) from SFV DNA replicons [46]. Mice immunized with SFV-BoNT/A Hc and SFV-GM-CSF DNA replicons prolonged the survival of mice significantly after challenges with BoNT/A. In another approach, the BoNT/E and BoNT/F serotype Hc genes were expressed from a dual SFV DNA replicon showing resistance against challenges with a BoNT/E and BoNT/F mixture in mice [47]. Moreover, SFV DNA replicons have been engineered for the expression of the tetravalent BoNT Hc genes for the A, B, E, and F serotypes and the tetanus neurotoxin (TeNT) [48]. Initially, the SFV-TeNT DNA replicon provided strong antibody responses and protection in BALB/c mice [48]. Furthermore, the SFV-TeNT DNA replicon was combined with the BoNT serotypes in a pentavalent DNA replicon vaccine, which induced specific antibody responses and provided protection against the five targets in mice [48]. Related to anthrax, SFV DNA replicons have been utilized for the expression of the anthrax protective antigen (PA) [49]. Both SFV-PA replicons and viral particles elicited stronger PA-specific immune responses than conventional plasmid DNA vectors, resulting in potent protection against challenges with the *Bacillus anthracis* A16R strain in mice. In the context of protozoans, SFV DNA replicon-based expression of the nucleoside triphosphate hydrolase-II of *Toxoplasma gondii* (TgNTPase-II) provided partial protection against acute infection and toxoplasmosis in mice [50].

In the context of parasites, the fusion protein of the SP15 protein from the insect *Phlebotomus papatasi* and the Leishmania major stress inducible protein 1 (PpSP15-LmSTI1) was expressed from SFV RNA replicons, SFV DNA replicons, and conventional plasmid DNA vectors [51]. In this case, superior expression was observed from SFV RNA replicons in comparison to DNA replicons or conventional DNA plasmids, suggesting that SFV RNA replicons are preferred for further vaccine development against leishmaniasis. 

## 4. DNA Replicons and Cancer

DNA replicons have been frequently evaluated for cancer therapeutics and vaccines (Table 2). For example, human papilloma virus (HPV), due to its role in cervical cancer, has been targeted by expression of the HPV type 16 E7 protein fused to the *M. tuberculosis* heat shock protein 70 (Hsp70) from SFV DNA replicons [52]. The fusion construct elicited superior E7-specific T-cell-mediated immune responses and exhibited superior potency against established metastatic tumors in mice compared to SFV-HPV-16 E7 alone [52]. Moreover, SFV DNA replicon-based expression of the HPV E6 and E7 antigens demonstrated efficient anti-tumor activity after intradermal administration, resulting in 85% of treated mice becoming tumor-free [53]. In another approach, the antiapoptotic BCL-xL gene was co-expressed with the HPV-E7 gene from an SFV DNA replicon to delay cell death in transfected dendritic cells [54]. It generated significantly higher E7-specific CD8^+^ T-cell-mediated immune responses and enhanced the antitumor activity in mice.

In the case of breast cancer, intramuscular administration of SIN DNA replicons expressing the HER2/neu gene generated robust immune responses and reduced tumor incidence and tumor mass in mice [55]. Moreover, intradermal immunization was highly efficient requiring 80% less SIN DNA replicon to achieve the same efficacy as seen for intramuscular injection [55]. The immunization also provided protection against spontaneous breast tumors and reduction in metastases developed from HER2/neu tumors. Moreover, mice with A2L2 mammary fat pad tumors were subjected to combination therapy with SIN-neu DNA replicons and doxorubicin or paclitaxel resulting in significant tumor growth reduction [56]. In contrast, neither SIN-neu DNA replicon treatment nor chemotherapy alone was efficient. Furthermore, SIN-neu DNA replicons were evaluated in mice in a solid mammary tumor model and a lung metastasis model [57]. Tumor growth was significantly inhibited in mice, which had been immunized with SIN-neu DNA replicons prior to challenges with A2L2 tumors. The same findings were observed after immunization with an Ad-neu vector [57]. In contrast, immunization with either SIN-neu DNA replicons or Ad-neu particles 2 days after tumor cell administration was inefficient. However, a prime immunization with SIN-neu DNA replicons followed by a booster immunization with Ad-neu particles significantly prolonged survival in mice.

In the context of melanoma, SIN DNA replicons expressing the self/tumor antigen tyrosine-related protein-1 (TRP1) in mice demonstrated that, unlike conventional DNA vaccines, DNA replicon-based vaccines can break tolerance and provide immunity against melanomas [58]. Double-stranded RNA formation was critical and DNA replicons were responsible for the activation of innate immune pathways as a means of improving the efficacy of DNA-based immunization. In another study on melanoma, SIN DNA replicon-based expression of the cell adhesion molecule (MCAM/MUC18) resulted in protection of immunized mice against challenges with B16F10 melanoma cells in both primary and metastatic mouse tumor models [59]. The synergistic effect of co-immunization of an SFV DNA replicon expressing the 1–4 domains of murine vascular epidermal growth factor receptor-2 (VEGFR2) and IL-12 and another SFV DNA replicon expressing the survivin and β-hCG antigens has been evaluated in a B16 mouse melanoma model [60]. Enhanced humoral and cellular immune responses and prolonged survival were obtained after co-administration of the two SFV DNA replicons. In the context of flaviviruses, KUN-based DNA replicons have demonstrated stable expression of the β-galactosidase (β-gal) reporter gene due to the noncytopathic nature of the KUN vector [61]. In vivo applications showed that intranasal inoculation of KUN- β-gal DNA replicons resulted in expression of β-gal in the mouse lung epithelium for at least 8 weeks. KUN DNA replicons expressing the OVA(257–264)(SIINFEKL) epitope have been encapsulated in lipoplex formulations engrafted with a flagellin-derived peptide (9Flg) for antigen-presenting cell (APC) targeting [63]. Intravenous administration of KUN-SIINFEKL DNA-lipoplexes elicited strong antigen-specific T-cell responses and induced significantly enhanced antitumor responses in a B16-OVA melanoma mouse model compared to conventional DNA plasmids. 

Related to brain tumors, co-expression of human gp100 and mouse interleukin-18 (IL-18) from SIN DNA replicons in mice with implanted B16-gp100 brain tumors showed both protective and therapeutic efficacy [62]. Both CD4^+^/CD8^+^ T cells and IFN-γ mediated the antitumor and protective effects, and the survival rate was significantly improved in mice with implanted B16 tumors. 

## 5. Comparison to Conventional DNA Vectors, RNA Replicons and Viral Particles

Comparative studies between DNA replicon-based vaccines and conventional DNA vaccines have indicated that significantly lower doses of DNA replicons are needed to elicit robust immune responses [35,37]. For example, robust antibody responses and protection against challenges with HSV were achieved with 100–1000-fold lower doses of SIN DNA replicons compared to conventional DNA plasmids [33]. Doses as low as 10 ng of DNA replicons were sufficient to elicit strong immune responses in mice [33]. In the case of HPV vaccine development, a dose of 50 ng of SFV DNA replicon generated complete tumor regression on 85% of mice [54]. In the context of HIV vaccines, only 3.2 ng of the SFV DNA replicon (DREP.HIVconsv) vaccine was needed to elicit HIV-1 specific CD8^+^ T-cell responses compared to 1 µg of conventional DNA vaccine, equivalent to a 625-fold higher molar dose [37]. In a study in rhesus macaques, it was shown that SFV DNA replicon doses could be reduced by 20-fold compared to conventional DNA vectors [64]. This overall trend of reduced doses decreases the demands on large-scale production of GMP grade vaccines and also permits the use of lower doses in humans, which might reduce both the severity and frequency of serious adverse events. An interesting observation was made from a comparative study of SFV DNA replicon (pSFVC1.5) and conventional DNA (pcDC) vectors expressing the pseudorabies virus (PrV) glycoprotein C (gC) [65]. Although both pSFVC1.5 and pcDC induced neutralizing antibodies in BALB/c mice, the levels were relatively lower for the DNA replicon-based immunization. However, the pSFVC1.5 vaccination provided 100% protection, whereas it was only 62.5% for pcDC. Furthermore, it was demonstrated that pSFVC1.5 elicited stronger lymphocyte proliferative responses and higher levels of interferon-γ (IFN-γ) indicating that pSFVC1.5 elicited an enhanced Th1-biased immune response. An important finding, as described above, was that SIN-TRP1 DNA replicon immunization managed to break tolerance and provided immunity against melanoma, which was not the case for immunization with conventional DNA-based vaccines [58]. The stronger immune responses obtained after immunization with SIN-p85 DNA replicons, which generated long-term protection against *M. tuberculosis* [43] was not achieved for conventional DNA vaccines.

The feasibility of DNA replicon-based vaccines has been verified by direct comparison not only to conventional DNA plasmid-based vaccines but also to recombinant viral particles and RNA replicons. For example, the medium (M) and small (S) gene segments of the envelope glycoproteins or nucleocapsid protein of the Seoul virus (SEOV), belonging to the hantavirus family, were compared for expression from a conventional DNA vector, a SIN DNA replicon, and recombinant SIN particles in Syrian hamsters [66]. All vector systems induced SEOV-specific immune responses and provided protection against SEOV challenges with the most consistent protection after immunization with the conventional DNA vector. However, generally recombinant viral particles have been highly successful in eliciting strong immune responses, providing protection against challenges with pathogenic infectious agents, and substantial tumor regression and eradication, as has previously been described [17]. Moreover, recombinant viral particles have been frequently subjected to clinical trials [17]. 

In the context of mRNA-based vaccines, there is no doubt that recent success seen for vaccines developed against SARS-CoV-2 leading to emergency use authorization and global mass vaccinations has been superior to any DNA vaccine development so far [67]. Related to RNA replicons, for example, immunization of mice with 0.1 µg of an SFV-LacZ RNA replicon elicited antigen-specific antibody and provided protection against tumor challenges [68]. Although the in vitro antigen production levels for RNA replicons were not significantly higher compared to those seen for conventional DNA vaccines, the enhanced tumor killing seen in vivo could be associated with the induced caspase-dependent apoptotic cell death. In another study, the RNA replicon-based SIN-Rab-G vaccine elicited similar cellular and humoral IgG responses compared to a conventional rabies DNA vaccine [69]. Immunization with 10 µg of SIN-Rab-G DNA replicon provided protection against challenges with the rabies CVS strain. Moreover, nanoparticle encapsulation of VEE RNA replicons (RNA/LNPs) has been compared to recombinant VEE, particles, naked RNA replicons, VEE DNA replicons, and conventional plasmid DNA by monitoring Luc reporter expression [70]. Similar Luc levels were detected for RNA/LNPs and VEE particles. In contrast, expression levels were significantly lower for naked RNA replicons, DNA replicons, and conventional plasmid DNA. Similarly, immune responses against the respiratory syncytial virus fusion protein (RSV-F) were much higher for RNA/LNPs and VEE particles than for naked RNA replicons, DNA replicons, and conventional plasmid DNA. Furthermore, cotton rats immunized with RNA/LNPs, RNA replicons, VEE particles, and an RSV-F subunit vaccine were all protected against RSV challenges, although the RSV-F subunit vaccine formulated with alum as an adjuvant showed the highest potency. 

## 6. Conclusions

DNA replicons have demonstrated protective and therapeutic efficacy in animal models for infectious diseases and various cancers, as summarized in Table 1 and Table 2. It has also been established that both immune responses and protection against infectious agents or tumor cells are at least equivalent and, in many cases, superior to those seen for conventional DNA vaccines. Most importantly, robust immunogenicity and therapeutic efficacy have been achieved with 100–1000-fold lower doses of DNA replicons compared to conventional DNA vaccines. Importantly, lower doses relate to potentially reduced serious adverse events caused by vaccination and also a more cost-effective large-scale production of vaccines. In comparison to vaccines based on RNA replicons or recombinant viral particles, the general impression is that especially viral particles, but also to some extent RNA replicons show superior immune responses and prophylactic and therapeutic efficacy compared to DNA replicons. Additionally, both recombinant viral particles and RNA replicons have been subjected to clinical trials. For example, the VSV-based VSV-ZEBOV vaccine provided excellent protection in phase III clinical evaluations [71,72] and was recently approved by the FDA [73]. Moreover, an LNP formulation of the SARS-CoV-2 vaccine based on a VEE RNA replicon has been evaluated in a phase I clinical trial [74]. Obviously, the rapid development and approval of mRNA-based SARS-CoV-2 vaccines have demonstrated the feasibility of this approach.

Aspects of superiority of using DNA replicons are their easy, inexpensive, and rapid handling for large-scale production. In comparison to the degradation-sensitive RNA replicons, the double-stranded structure of DNA replicons is stable, which allows easier handling, transport, and storage of DNA-based vaccines. Furthermore, it has been demonstrated that DNA plasmids are not integrated into the host genome at any level of concern. In comparison to viral particles, there is no safety risk of using DNA-based material. However, as indicated in comparative studies, expression levels from DNA replicons are often inferior to those obtained from RNA replicons or recombinant viral particles, which might be related to the requirement of delivery of DNA to the cell nucleus. Generally, delivery improvement through LNP and lipoplex formulations as well as improved nuclear translocation by the introduction of nuclear localization signals need to be explored to be able to provide more efficient DNA replicons for future gene therapy and vaccine exploration.

## Figures and Tables

**Figure 1 pharmaceutics-15-00947-f001:**
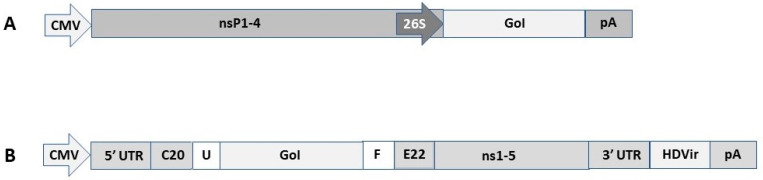
Schematic presentation of DNA replicon expression vectors. (**A**). Alphavirus-based DNA replicon represented by a Semliki Forest virus (SFV) vector. CMV, cytomegalovirus promoter; nsP1-4, non-structural protein genes forming the replicase complex; 26S, SFV 26S subgenomic promoter; GoI, gene of interest; pA, polyadenylation signal. (**B**). Flavivirus-based DNA replicon represented by a Kunjin virus (KUN) vector. 5′ UTR, 5′ untranslated region; C20, the first 20 amino acids of KUN C protein; U, mouse ubiquitin sequence; F, Foot-and-mouth disease virus 2A autoprotease; E22, the last 22 amino acids of KUN E protein; 3′ UTR, 3′ untranslated region; HDVr, Hepatitis delta virus ribozyme.

**Figure 2 pharmaceutics-15-00947-f002:**
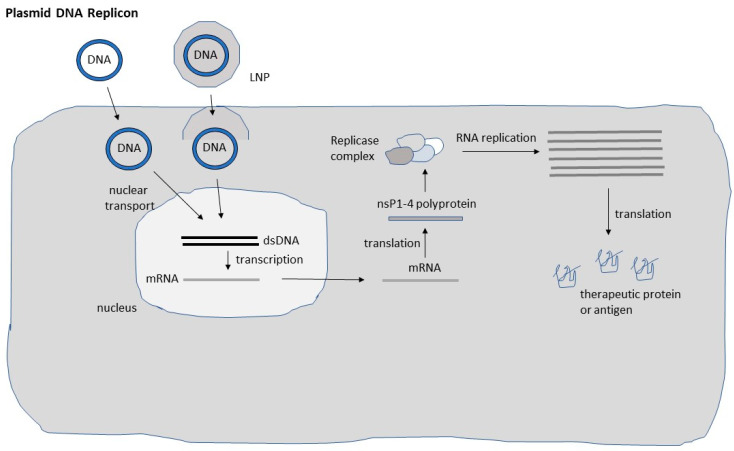
Schematic presentation of DNA replicon systems. Both naked and lipid nanoparticle (LNP) encapsulate DNA replicons have been used. DNA replicon delivered to the nucleus is in vivo transcribed and mRNA translocated to the cytoplasm for translation of the non-structural (nsP1-4) polyprotein, which is processed to individual nsP1-4 proteins forming the replicase complex. The replicase complex is responsible for massive self-replication of RNA, leading to the translation of high levels of recombinant protein or antigen.

**Table 1 pharmaceutics-15-00947-t001:** Examples of DNA replicon-based vaccines against infectious agents and toxins.

Disease	DNA Vector/Target	μg DNA	Findings
**Viral**			
HSV-1	SIN/HSV-1 gB	0.01–3	Protection in mice against HSV-1 after single injection [33]
MV	SIN/MV-H	100	100% protection against MV in cotton rats [34]
MV	SIN/MV-H-Fdu	100	Protection only after booster with live MV vaccine in rats [34]
BVDV	SFV/BVDV p80	100	CTL and CMI responses against BVDV in mice [35]
CSFV	SFV/CSFV-E2 + rAdV-E2	100	Heterologous prime-boost strategy superior in pig model [36]
HIV	SFV/HIV Env, GagPolNef	0.2	Greatly enhanced immune responses after MVA booster [37]
HCV	SFV/HCV C, E1, E2, NS2/3	0.5–50	CTL and humoral responses after MVA booster in mice [38]
EBOV	SFV/EBOV GP, VP40	5	Binding and neutralizing Abs in mice [39]
EBOV	SFV/EBOV GP, VP40	10	Superior humoral, cellular responses after co-injection [40]
VEE	VEE/V4020 genome	100	Protection against VEE in mice [41]
VEE	VEE/V4020 genome	100	Protection against VEE in rhesus macaques [42]
COVID-19	SFV/SARS-CoV-2 S	10	Neutralizing Abs, superior IgG Abs in mice after 1 injection [43]
COVID-19	SFV/SARS-CoV-2 S^ecto^	10	Neutralizing Abs in mice [43]
**Bacterial**			
TB	SIN/*M. tuberculosis* p85	0.5–50	Specific Ab responses, protection against challenges in mice [44]
TB	VEE/Acr-Ag85B fusion	20	Inhibition of bacterial growth in lungs and spleen of mice [45]
Botulism	SFV/BoNT/A Hc, GM-CSF	100	Prolonged survival after BoNT/A challenges in mice [46]
Botulism	SFV/BoNT/E, BoNT/F	100	Protection against challenge with BoNT/E-BoNT/F mixture [47]
Botulism	SFV/BoNT/A, B, E, F	100	Protection against 4 BoNT serotypes in mice [48]
Tetanus	SFV/TeNT	100	Protection against TeNT in mice [48]
Anthrax	SFV/anthrax PA	100	Protection against *B. anthracis* A16R strain in mice [49]
**Protozoan**			
TP	SFV/Tg-NPase II	100	Protection against acute infection, toxoplasmosis in mice [50]
**Parasites**			
LD	SFV/PpSP15-LmST11	0.5–2	Superior expression from RNA than DNA replicons [51]

Abs, antibodies; BoNT, botulinum neurotoxin; BVDV, bovine viral diarrhea virus; CMI, cell-mediated immune; CSFV, classical swine fever virus; CTL, cytotoxic T-cell; EBOV, Ebola virus; HCV, hepatitis C virus; HIV, human immunodeficiency virus; HSV-1 gB, herpes simplex virus-1 glycoprotein B; LD, leishmaniasis disease; MV, measles virus; MVA, modified poxvirus Ankara strain; MV-H, MV hemagglutinin, MV-H-Fdu, MV hemagglutinin-fusion protein; rAdV, recombinant adenovirus; SFV, Semliki Forest virus; SIN, Sindbis virus; TB, tuberculosis; TP, toxoplasmosis.

**Table 2 pharmaceutics-15-00947-t002:** Examples of DNA replicon-based cancer therapy and vaccines.

Cancer	DNA Vector/Target	μg DNA	Findings
Cervical	SFV/HPV E7-Hsp70	2	Potential antitumor activity in metastases [52]
Cervical	SFV/HPV E6-E7	0.05	85% of treated mice becoming tumor-free [53]
Cervical	SFV/HPV E7-BCL-xL	2–20	Higher immunogenicity, enhanced antitumor activity in mice [54]
Breast	SFV/HER2/neu	100	Tumor regression, protection against tumors in mice [55]
Breast			
Breast	SIN/neu + Dox/Pac	100	Substantial tumor regression with Dox or Pac in mice [56]
	SIN/neu + Ad-neu	100	Booster vaccination with Ad-neu prolonged survival in mice [57]
Melanoma	SIN/TRP1	3	Break of tolerance, immunity against melanomas in mice [58]
Melanoma	SIN/MCAM/MUC18	50	Protection against B16F10 melanoma challenges in mice [59]
Melanoma	SFV/VEGFR2/IL-12 + Survivin/β-hCG Ag	50	Enhanced humoral, cellular immune responses, prolonged survival after co-administration of DNA replicons in mice [60]
Melanoma	KUN/SIIINFEKL-LPX	25	Enhanced antitumor activity in mice [61]
Brain	SIN/gp100/IL-18	100	Protective, therapeutic effects, prolonged survival in mice [62]

Ad, adenovirus; Dox, doxorubicin; HPV, human papilloma virus; Hsp70, *M. tuberculosis* heat shock protein 70; IL. interleukin; MCAM/MUC18, melanoma cell adhesion molecule; Pac, paclitaxel; TRP-1, tyrosine-related protein-1; VEGFR2, vascular endothelial growth factor receptor-2.

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
