# Peer review of "Application of DNA Replicons in Gene Therapy and Vaccine Development"

_pharmaceutics, 2023, doi:10.3390/pharmaceutics15030947_

Round 1

Reviewer 1 Report

DNA replicon vectors based on RNA viruses were first described in 1996. So this is not new technology. While the article cites many publications that highlight the potential of this technology for vaccines and cancer therapy, only about a third of the primary literature cited for these applications are recent (in the last 5 years). This rather points to the technology as being outdated. Additionally, none of the more current literature referenced reported a direct comparison with current benchmark technologies in the field, e.g. mRNA vaccines (currently used against viruses and as personalized cancer vaccines). The main point and bulk of the article was to argue that this technology is better than plasmid DNA technology, e.g. for vaccines. But mRNA technology has potentially already surpassed plasmid DNA technology for vaccines, including cancer vaccines. A quick look in clinicaltrials.gov with the search terms “vaccine” and “plasmid DNA” returns 178 studies, whereas the search terms “vaccine” and “mRNA” returns 425. Yes, due to the recent pandemic, mRNA technology has overshadowed technologies such as DNA replicon vectors. But rather than ignore this current trend, the author could choose to address "the competition" directly and objectively highlight if and where DNA replicon technology has a place in vaccinology, cancer therapy and gene therapy. This would enhance the timeliness and relevance of the manuscript.

The two sentences starting at line 57 on page 2 copied below. The highlighted words should read in vitro. There is no T7 RNA polymerase in the mammalian cell, and there would be no need to electroporate or transfect an in vivo-transcribed RNA.

“The self-replicating vector systems contain the genes for the viral non-structural proteins and the gene of interest (GoI), which are in vivo transcribed from an SP6 or T7 RNA polymerase promoter [20]. Electroporation or transfection of in vivo transcribed RNA leads to extensive RNA replication in the cytoplasm of host cells and when helper RNA containing the viral structural genes is co-administered recombinant viral particles are generated.”

Author Response

DNA replicon vectors based on RNA viruses were first described in 1996. So this is not new technology. While the article cites many publications that highlight the potential of this technology for vaccines and cancer therapy, only about a third of the primary literature cited for these applications are recent (in the last 5 years). This rather points to the technology as being outdated. Additionally, none of the more current literature referenced reported a direct comparison with current benchmark technologies in the field, e.g. mRNA vaccines (currently used against viruses and as personalized cancer vaccines). The main point and bulk of the article was to argue that this technology is better than plasmid DNA technology, e.g. for vaccines. But mRNA technology has potentially already surpassed plasmid DNA technology for vaccines, including cancer vaccines. A quick look in clinicaltrials.gov with the search terms “vaccine” and “plasmid DNA” returns 178 studies, whereas the search terms “vaccine” and “mRNA” returns 425. Yes, due to the recent pandemic, mRNA technology has overshadowed technologies such as DNA replicon vectors. But rather than ignore this current trend, the author could choose to address "the competition" directly and objectively highlight if and where DNA replicon technology has a place in vaccinology, cancer therapy and gene therapy. This would enhance the timeliness and relevance of the manuscript.

Response: I respectfully disagree with the reviewer that the DNA replicon-based approach is outdated. In contrast, I believe that it has not been properly utilized yet and there is great potential as demonstrated by the work carried out by Szurgot and co-workers (Ref 43) on DNA-replicon-based vaccines against SARS-CoV-2. I am a bit surprised that the reviewer feels that the superiority of mRNA-based vaccines to DNA-based approaches is not clearly pointed out. I fully agree that mRNA-based vaccines have been applied more frequently than DNA-based approaches, especially with the development of mRNA-based vaccines against SARS-CoV-2. In section 5. comparisons between DNA replicon-based and conventional DNA vaccines, RNA replicons (mRNA vaccines) and recombinant viral particles are made. Perhaps I have too strongly restricted any further expansion on mRNA-based vaccines as these have been described in several other reviews (see Aljabali et al 2023 Drug Discov. Today 28, 103458; Lundstrom 2020 Int. J. Mol. Sci. 21, 5130; Lundstrom 2021 Vaccines 9, 1187). Although in the Conclusions section is stated “In comparison to vaccines based on RNA replicons or recombinant viral particles, the general impression is that especially viral particles, but also to some extent RNA replicons show superior immune responses and prophylactic and therapeutic efficacy compared to DNA replicons”. In any case, I have expanded this section and also included comments on the mRNA vaccine approach in the Conclusions section.

The two sentences starting at line 57 on page 2 copied below. The highlighted words should read in vitro. There is no T7 RNA polymerase in the mammalian cell, and there would be no need to electroporate or transfect an in vivo-transcribed RNA.

“The self-replicating vector systems contain the genes for the viral non-structural proteins and the gene of interest (GoI), which are in vivo transcribed from an SP6 or T7 RNA polymerase promoter [20]. Electroporation or transfection of in vivo transcribed RNA leads to extensive RNA replication in the cytoplasm of host cells and when helper RNA containing the viral structural genes is co-administered recombinant viral particles are generated.”

Response: Sorry for the mistake. It has now been corrected.

Reviewer 2 Report

The current review gives a comprehensive overview of DNA replicons. Just one point: Figure 2 doesn’t add much information in the current status (just showing transcription/replication/translation is a bit too basic. I recommend adding some info on the replicase complex and amend the legend accordingly. This would be helpful to attract a broader readership.

Author Response

The current review gives a comprehensive overview of DNA replicons. Just one point: Figure 2 doesn’t add much information in the current status (just showing transcription/replication/translation is a bit too basic. I recommend adding some info on the replicase complex and amend the legend accordingly. This would be helpful to attract a broader readership.

Response: Additional info has been added to Fig. 2 and the figure legend.

Reviewer 3 Report

The paper is well written and sound.

I have only minor remark:

There are few sentences that need to be rewritten to make them more clear:

Lines 46-48 and lines 59-62 sentences are not completely clear.

I think some more detailed explanation on the mechanism of DNA replicons would and some addition value to the paper.

Author Response

The paper is well written and sound.

I have only minor remark:

There are few sentences that need to be rewritten to make them more clear:

Lines 46-48 and lines 59-62 sentences are not completely clear

Response: I have looked at the sentences on lines 46-48 and 59-62 and could not figure out what is unclear except in the latter case the mistake of describing “in vivo” transcription, which obviously should be “in vitro” transcription. Let me know if there anything else that is not clear.

I think some more detailed explanation on the mechanism of DNA replicons would and some addition value to the paper.

Response: Additional info has been added to Fig. 2 and the figure legend.

Round 2

Reviewer 1 Report

·       The author has not addressed my major point. The author says he has expanded on Section 5 and conclusions, but there are no changes in these sections. They are exactly the same.

·       Likewise, the author makes references to alterations in Figure 2 and accompanying legend (in response to other reviewers) that are non-existent in the revised manuscript. The author should make sure that the correct revised manuscript has been uploaded.

·       The author believes that DNA replicon technology has not reached its full potential and he is potentially correct. Unfortunately, the main takeaways from this review are that DNA replicons can outperform conventional plasmid-based vaccines (and potentially some viral-based vaccines as well), but RNA-based vaccines are superior. Therefore, the reader is left wondering: if RNA-based platforms are superior (as per the author’s own words), then what is the utility/relevance of a review about DNA replicon technology? The author needs to make a stronger argument in the conclusions for DNA replicon technology against the current landscape of RNA-based vaccine platforms. Essentially: if I were developing a vaccine, how would you convince me to choose for a DNA replicon platform over an mRNA or RNA replicon platform? Is there evidence in side-by-side comparisons that DNA replicon technology can at least match their RNA counterparts in efficacy?

Author Response

Dear Reviewer,

It seems that you have received the original and not the revised version of the manuscript. I have uploaded the right version again and I hope this is ok. My apologies for this inconvenience.